# Functional CDKN2A assay identifies frequent deleterious alleles misclassified as variants of uncertain significance

Hirokazu Kimura[1], Raymond M Paranal[1,2], Neha Nanda[1], Laura D Wood[1,3], James R Eshleman[1,3,4], Ralph H Hruban[1,3], Michael G Goggins[1,3], Alison P Klein[1,3,4], The Familial Pancreatic Cancer Genome Sequencing Project, Nicholas J Roberts[1,3]*

[1]The Sol Goldman Pancreatic Cancer Research Center, Department of Pathology, Johns Hopkins University, Baltimore, United States; [2]Human Genetics Predoctoral Training Program, the McKusick-Nathans Institute of Genetic Medicine, The Johns Hopkins University School of Medicine, Baltimore, United States; [3]Department of Oncology, The Johns Hopkins University School of Medicine, Baltimore, United States; [4]Department of Epidemiology, The Johns Hopkins University Bloomberg School of Public Health, Baltimore, United States

**Abstract** Pathogenic germline *CDKN2A* variants are associated with an increased risk of pancreatic ductal adenocarcinoma (PDAC). *CDKN2A* variants of uncertain significance (VUSs) are reported in up to 4.3% of patients with PDAC and result in significant uncertainty for patients and their family members as an unknown fraction are functionally deleterious, and therefore, likely pathogenic. Functional characterization of *CDKN2A* VUSs is needed to reclassify variants and inform clinical management. Twenty-nine germline *CDKN2A* VUSs previously reported in patients with PDAC or in ClinVar were evaluated using a validated in vitro cell proliferation assay. Twelve of the 29 *CDKN2A* VUSs were functionally deleterious (11 VUSs) or potentially functionally deleterious (1 VUS) and were reclassified as likely pathogenic variants. Thus, over 40% of *CDKN2A* VUSs identified in patients with PDAC are functionally deleterious and likely pathogenic. When incorporating VUSs found to be functionally deleterious, and reclassified as likely pathogenic, the prevalence of pathogenic/likely pathogenic *CDKN2A* in patients with PDAC reported in the published literature is increased to up to 4.1% of patients, depending on family history. Therefore, *CDKN2A* VUSs may play a significant, unappreciated role in risk of pancreatic cancer. These findings have significant implications for the counselling and care of patients and their relatives.

*For correspondence:
nrobert8@jhmi.edu

Group author details:
The Familial Pancreatic Cancer Genome Sequencing Project See page 13

## Introduction

Pancreatic ductal adenocarcinoma (PDAC) is a lethal cancer with a median survival of less than 6 months and a 5-year survival rate of only 10% (*Siegel et al., 2020*). Surveillance of high-risk individuals (HRIs), such as people with a pathogenic germline variant in a pancreatic cancer susceptibility gene, for example, *ATM*, *BRCA1*, *BRCA2*, *CDKN2A*, *CPA1*, *CPB1*, *MLH1*, *MSH2*, *MSH6*, *PALB2*, *PMS2*, *PRSS1*, and *STK11*, has the potential to reduce mortality through detection of early, potentially curable, PDAC and its precursor lesions (*Canto et al., 2018*; *Vasen et al., 2016*). The International Cancer of the Pancreas Screening (CAPS) Consortium trials found that surveillance identified either PDAC or a high-grade precursor lesion in 6.8% of HRIs, including relatives of patients with familial pancreatic cancer and individuals with a pathogenic germline variant in a pancreatic cancer susceptibility gene, during a median follow-up 5.6 years (24 of 354 individuals) (*Canto et al., 2013*; *Goggins et al., 2020*; *Canto et al., 2018*). Similarly, in a recent screening study of 178 HRIs with

a germline pathogenic *CDKN2A* variant, 13 individuals with PDAC were identified (7.3%) during a median follow-up 53 months (*Vasen et al., 2016*). Importantly, the overall resection rate and 5-year survival rate was 75% and 24%, respectively (*Vasen et al., 2016*). Furthermore, patients with PDAC and a pathogenic germline variant in a pancreatic cancer susceptibility gene may have cancers that are uniquely sensitive to specific anti-cancer therapies, as is the case for poly(ADP)-ribose polymerase inhibitors and immunotherapy (*Kaufman et al., 2015*; *Le et al., 2015*). Consequently, recent American Society of Clinical Oncology (ASCO) and National Comprehensive Cancer Network (NCCN) guidelines recommend germline testing of all pancreatic cancer patients and their at-risk first-degree relatives (*Goggins et al., 2020*; *Stoffel et al., 2019*).

The *CDKN2A* gene encodes two proteins, p16$^{INK4a}$ and p14$^{ARF}$. p16$^{INK4a}$ inhibits CDK4 activity, is a regulator of cell cycle progression, and a tumor suppressor gene (*Serrano et al., 1993*). Germline variants in *CDKN2A* that affect p16$^{INK4a}$ (hereafter referred to *CDKN2A* variants) can be classified as either pathogenic, benign, or variants of uncertain significance (VUSs) based on American College of Medical Genetics (ACMG) guidelines (*Richards et al., 2015*). Pathogenic germline *CDKN2A* variants have been identified in up to 3.3% patients with familial pancreatic cancer and up to 2.6% of patients with PDAC unselected for family history or without a family history of PDAC (*Brand et al., 2018*; *Chaffee et al., 2018*; *Hu et al., 2018*; *Kimura et al., 2021*; *Lowery et al., 2018*; *McWilliams et al., 2018*; *McWilliams et al., 2011*; *Roberts et al., 2016*; *Shindo et al., 2017*; *Singhi et al., 2019*; *Zhen et al., 2015*). Strikingly, individuals with a known pathogenic germline *CDKN2A* variant have up to a 12.3-fold increased risk of developing PDAC (*Hu et al., 2018*).

Germline *CDKN2A* VUSs are a common finding in patients undergoing germline genetic testing. *CDKN2A* VUSs, predominantly rare missense variants, are found in 2.9–4.3% of patients with PDAC, depending on family history (*Chaffee et al., 2018*; *McWilliams et al., 2018*; *Roberts et al., 2016*; *Shindo et al., 2017*; *Zhen et al., 2015*). Finding a germline *CDKN2A* VUS can be a cause of significant clinical uncertainty. For example, while individuals with a pathogenic germline *CDKN2A* variant are recommended for clinical surveillance, those with a germline *CDKN2A* VUS are often not included in surveillance studies unless they fulfill family history criteria (*Goggins et al., 2020*; *Stoffel et al., 2019*). Therefore, the functional characterization and reclassification of *CDKN2A* VUSs will inform disease surveillance and early detection efforts.

The effect of *CDKN2A* variants can be determined in vitro by assay of a specific molecular function of p16$^{INK4A}$, such as binding to CDK4/6, or by assay a broad cellular function, such as cell proliferation, cell viability, and cell cycle progression (*Kannengiesser et al., 2009*; *Miller et al., 2011*; *Ng et al., 2018*). The positive and negative predictive values for assays of broad cellular function are higher than those considering only CDK4/6 binding (*Miller et al., 2011*). For example, some *CDKN2A* variants, such as p.Gly35Ala, p.Gly67Arg, p.Glu69Gly, and p.Arg87Trp, bind to CDK4 comparably to wild type (WT) p16$^{INK4a}$, but do not inhibit cell proliferation, indicating that additional functions not assessed in vitro CDK4 binding are important for cellular function (*Kannengiesser et al., 2009*).

**Table 1.** *CDKN2A* variants assayed.

| Classification[a] | Variant[b] |
|---|---|
| Pathogenic | p.Leu16Arg, p.Arg24Pro, p.Met53Ile, p.Gly101Trp, p.Val126Asp |
| Likely pathogenic | p.Ile49Thr, p.Leu78Hisfs*41, p.His83Tyr, p.Asp84Asn |
| Likely benign | p.Ala57Val, p.Ala100Ser, p.His123Gln, p.Ala127Ser, p.Arg144Cys |
| Benign | p.Ala148Thr |
| VUS | p.Met9Thr, p.Pro11Leu, p.Thr18Pro, p.Ala20Gly, p.Arg24Gln, p.Pro41Gln, p.Gln50Arg, p.Leu65Pro, p.His66Pro, p.His66Arg, p.Gly67Arg, p.Ala68Val, p.GluE69Gly, p.Asp74Ala, p.Asp74His, p. Asp84Ala, p.Gly89Asp, p.Arg99Gly, p.Gly101Arg, p.Ala109Pro, p.Gly111Ser, p.Gly122Val, p.Arg124Cys, p.Asp125His, p.Ala127Pro, p.Arg128Pro, p.Ala134Thr, p.Gly139Arg, and p.Ala143Thr |

a. Classification using American College of Medical Genetics (ACMG) variant classification guidelines.

b. Variant given as protein change with reference to NP_000068.1. Known pathogenic and benign variants used for benchmarking indicated in bold.

In this study, we used a validated in vitro cell proliferation assay and cell cycle analysis to determine the functional consequence of *CDKN2A* variants and reclassify 29 germline *CDKN2A* VUSs previously reported in patients with PDAC.

## Results

### *CDKN2A* variants assayed

The germline *CDKN2A* variants functionally characterized in this study are presented in *Table 1* and *Supplementary file 1*. Variants were classified using ACMG variant classification guidelines and given as amino acid change with reference to NP_000068.1 (*Richards et al., 2015*). We defined benchmark pathogenic variants as variants previously reported in patients with familial pancreatic cancer, PDAC, or reported in ClinVar as pathogenic or likely pathogenic. Benchmark pathogenic variants included: p.Leu16Arg, p.Arg24Pro, p.Ile49Thr, p.Met53Ile, p.Leu78Hisfs*41, p.His83Tyr, p.Asp84Asn, p.Gly101Trp, and p.Val126Asp (*Chaffee et al., 2018*; *Chang et al., 2016*; *Horn et al., 2021*; *Hu et al., 2018*; *McWilliams et al., 2018*; *Roberts et al., 2016*; *Zhen et al., 2015*). We defined benchmark benign variants as variants reported in ClinVar as benign or likely benign. Benchmark benign variants included: p.Ala57Val, p.Ala100Ser, p.His123Gln, p.Ala127Ser, p.Arg144Cys, and p.Ala148Thr (*McWilliams et al., 2018*; *Roberts et al., 2016*). Germline *CDKN2A* VUSs were selected as detailed in *Figure 1*. Briefly, 29 missense *CDKN2A* variants affecting p16[INK4A] that were previously reported in patients with familial pancreatic cancer, PDAC, or reported in ClinVar as VUSs were selected (*Chaffee et al., 2018*; *Hu et al., 2018*; *McWilliams et al., 2018*; *Roberts et al., 2016*; *Shindo et al., 2017*; *Zhen et al., 2015*). These 29 VUSs included: p.Met9Thr, p.Pro11Leu, p.Thr18Pro, p.Ala20Gly, p.Arg24Gln, p.Pro41Gln, p.Gln50Arg, p.Leu65Pro, p.His66Pro, p.His66Arg, p.Gly67Arg, p.Ala68Val, p.Glu69Gly, p.Asp74Ala, p.Asp74His, p. Asp84Ala, p.Gly89Asp, p.Arg99Gly, p.Gly101Arg, p.Ala109Pro, p.Gly111Ser, p.Gly122Val, p.Arg124Cys, p.Asp125His, p.Ala127Pro, p.Arg128Pro, p.Ala134Thr, p.Gly139Arg, and p.Ala143Thr.

### Validation of in vitro functional assay

PANC-1 (CRL-1469) is a human PDAC cell line with a homozygous deletion of *CDKN2A* (*Caldas et al., 1994*). Consistent with previous reports, CDKN2A (p16[INK4A]) was not detectable in PANC-1 cells by western blot (*Figure 2A*). Stable expression of WT CDKN2A in PANC-1 cells through lentiviral transduction resulted in a significant reduction in cell proliferation compared to empty vector control (p-value < 0.0001) (*Figure 2A and B*). Conversely, stable expression of three known pathogenic, and therefore functionally deleterious, *CDKN2A* variants, p. Leu78Hisfs*41, p.Gly101Trp, or p.Val126Asp, in PANC-1 cells resulted in no reduction in cell proliferation compared to the empty vector control (*Figure 2A and B*).

To establish benchmarks for the functional interpretation of *CDKN2A* VUSs, we next determined cell proliferation values for an additional six known pathogenic and six known benign variants in our assay. Known pathogenic and benign variants are likely functionally deleterious and functionally neutral, respectively, in our assay and were used as benchmarks for interpretation of *CDKN2A* VUSs. The mean cell proliferation value for all pathogenic variants assayed was 0.90 (range: 0.84–1.03), while the mean cell proliferation value for all benign variants assayed was 0.26 (range: 0.14–0.48) (*Figure 3*). Importantly, there was clear demarcation between the mean cell proliferations values of assayed benchmark pathogenic and benign variants. Therefore, we set the following thresholds to characterize assayed VUSs. VUSs were characterized as either functionally deleterious or functionally neutral if they had mean cell proliferation values of >0.81 or <0.44, respectively, based on the Z-score 95% confidence interval (CI). To classify assayed VUSs with mean cell proliferations values ≥0.44 but ≤0.81, we used a cutoff of 0.66, the midpoint between the lowest mean cell proliferation value for a pathogenic variant and the highest mean cell proliferation value for a benign variant, as area under the receiver operating characteristic (ROC) curve was 1, indicating that our assay could distinguish between pathogenic and benign variants with >99% sensitivity and >99% specificity. Therefore, assayed VUSs with a mean cell proliferation value >0.66 and ≤0.81 were classified as potentially functionally deleterious, while VUSs with a mean cell proliferation value >0.44 than ≤0.66 classified as potentially functionally neutral (*Figure 3*).

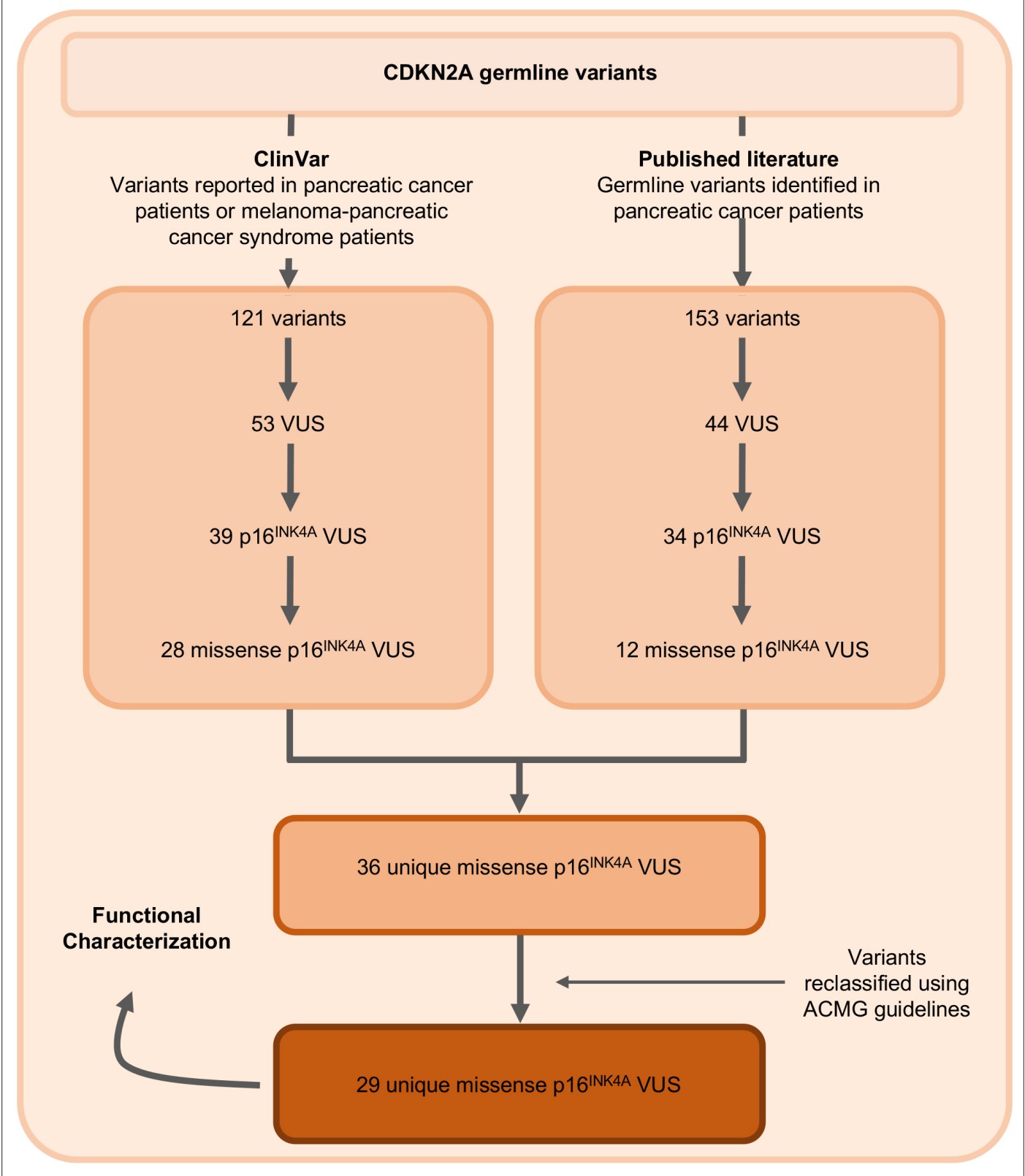

**Figure 1.** Identification of germline *CDKN2A* variant of uncertain significance (VUS) for functional characterization. Workflow to identify the 29 germline *CDKN2A* VUS selected for characterization in our functional assay. CDKN2A VUSs were classified using American College of Medical Genetics (ACMG) guidelines and either identified in patients with pancreatic ductal adenocarcinoma (PDAC) from published literature or in ClinVar.

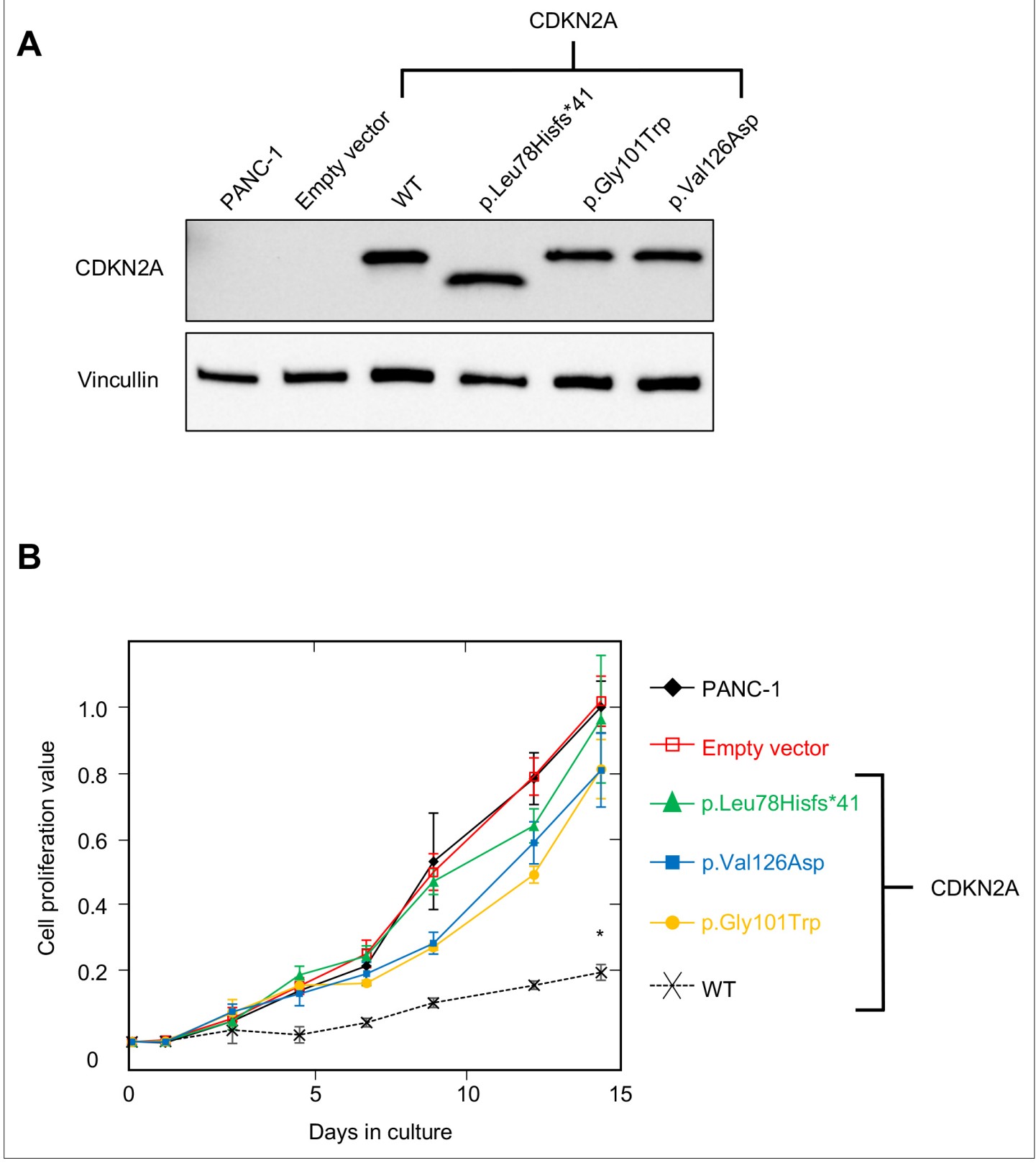

**Figure 2.** Development and validation of functional assay. (**A**) Western blot of whole cell lysates from PANC-1 cells and PANC-1 cells stably expressing wild type CDKN2A (p16INK4A) or selected pathogenic variants using anti-CDKN2A and anti-vinculin antibodies as a loading control. No expression of CDKN2A (p16INK4A) detected in PANC-1 cells or PANC-1 cell stably expressing empty vector. CDKN2A (p16INK4A) expression detected in PANC-1 cells stably expressing pathogenic variants. (**B**) Growth of PANC-1 cell and PANC-1 cells stably expressing wild type CDKN2A cDNA (p16INK4A) or selected

*Figure 2 continued on next page*

*Figure 2 continued*

pathogenic variants over 14 days in culture. Cell proliferation values are shown as mean (n = 3) ± s.d. normalized to PANC-1 cell stably expressed with empty vector. Significant growth inhibition in PANC-1 cell stably expressing wild type CDKN2A (p16$^{INK4A}$). *p < 0.01. Original files of the full raw unedited blots and the uncropped blots with the relevant CDKN2A and vinculin bands presented in *Figure 2—source data 1*, *Figure 2—source data 2* and 3. For raw data in *Figure 2*, please refer to *Figure 2—source data 4*.

The online version of this article includes the following source data for figure 2:

**Source data 1.** Original file of the full raw unedited CDKN2A blot.

**Source data 2.** Original file of the full raw unedited vinculin blot.

**Source data 3.** The uncropped blots with the relevant CDKN2A and vinculin bands labelled.

**Source data 4.** Raw data in *Figure 2B*.

## Functional characterization and reclassification of *CDKN2A* VUS

We used our cell proliferation assay to functionally characterize and reclassify 29 germline *CDKN2A* VUSs. These VUSs were selected because they were either reported in previously published studies in patients with PDAC and/or were reported in ClinVar (*Figure 3*). Eleven of the 29 VUSs (37.9%) had cell proliferation scores above 0.81 and were classified as functionally deleterious, including p.Thr18Pro, p.Ala20Gly, p.Gln50Arg, p.His66Pro, p.Ala68Val, p.Asp74Ala,, p.Asp84Ala, p.Gly89Asp, p.Ala109Pro, p.Ala127Pro, and p.Arg128Pro. One VUS (3.4%), p.Leu65Pro, had a cell proliferation score between 0.66 and 0.81 and was classified as potentially functionally deleterious. Three VUSs (10.3%), p.Asp74His, p.Gly111Ser, and, p.Gly122Val, had a cell proliferation score between 0.66 and 0.44 and were classified as potentially functionally neutral. The remaining 14 VUSs (48.3%) had cell proliferation scores below 0.44 and were classified as functionally neutral, including p.Met9Thr, p.Pro11Leu, p.Arg24Gln, p.Pro41Gln, p.His66Arg, p.Gly67Arg, p.Glu69Gly, p.Arg99Gly, p.Gly101Arg, p.Arg124Cys, p.Asp125His, p.Ala134Thr, p.Gly139Arg, p.Ala143Thr. Taken together, 12 of the 29 germline *CDKN2A* variants previously classified as VUSs, representing over 40% of assayed VUSs, demonstrated aberrant function in our assay and would be reclassified as likely pathogenic based on ACMG guidelines that consider validated in vitro functional assays showing a deleterious effect for a variant as strong evidence to support pathogenicity (*Richards et al., 2015*).

## Assay of CDKN2A VUSs in additional PDAC cell lines

To determine if the PDAC cell line used in our assay affected variant functional characterizations, we assayed all *CDKN2A* variants (benchmark pathogenic, benchmark benign, and VUS) using two additional PDAC cell lines with homozygous deletions of *CDKN2A*: MIA PaCa-2 (ATCC, catalog no. CRL-1420) and AsPC-1 (ATCC, catalog no. CRL-1682) (*Yamato and Furukawa, 2001*). Western blot for CDKN2A did not detect expression of p16$^{INK4A}$ in either cell line (*Figure 3—figure supplement 1*). Importantly, there was clear demarcation between the mean cell proliferation values of assayed benchmark pathogenic and benign variants in both MIA PaCa-2 and AsPC-1 (*Figure 3—figure supplement 2A-B* and *Supplementary file 2*). In MIA PaCa-2 cells, 7 (24.1%) VUSs were characterized as functionally deleterious, 5 (17.2%) as potentially functionally deleterious, 3 (10.3%) as potentially functionally neutral, and 14 (48.3%) as functionally neutral based on their mean cell proliferation values (*Figure 3—figure supplement 2A* and *Supplementary file 2*). In AsPC-1 cells, 6 (20.7%) VUSs were characterized as functionally deleterious, 6 (20.7%) as potentially functionally deleterious, 8 (27.6%) as potentially functionally neutral, and 9 (31%) as functionally neutral based on their mean cell proliferation values (*Figure 3—figure supplement 2B* and *Supplementary file 2*). Importantly, there was classification agreement between VUS functional classifications across cell lines (*Supplementary file 2*). When considering mean cell proliferation values for variants across all cell lines, 11 VUSs (20.7%; p.Thr18Pro, p.Gln50Arg, p.Leu65Pro, p.His66Pro, p.Ala68Val, p.Asp74Ala, p.Asp84Ala, p.Gly89Asp, p.Ala109Pro, p.Ala127Pro, and p.Arg128Pro) were classified as functionally deleterious, 1 VUS (3.4%; p.Ala20Gly) was potentially functionally deleterious, 2 VUSs (6.9%; p.Asp74His and p.Gly122Val) were potentially functionally neutral, and 15 VUSs (51.7%; p.Met9Thr, p.Pro11Leu, p.Arg24Gln, p.Pro41Gln, p.His66Arg, p.Gly67Arg, p.Glu69Gly, p.Arg99Gly, p.Gly101Arg, p.Gly111Ser, p.Arg124Cys, p.Asp125His, p.Ala134Thr, p.Gly139Arg, and p.Ala143Thr) were functionally neutral (*Figure 3—figure supplement 3* and *Supplementary file 2*).

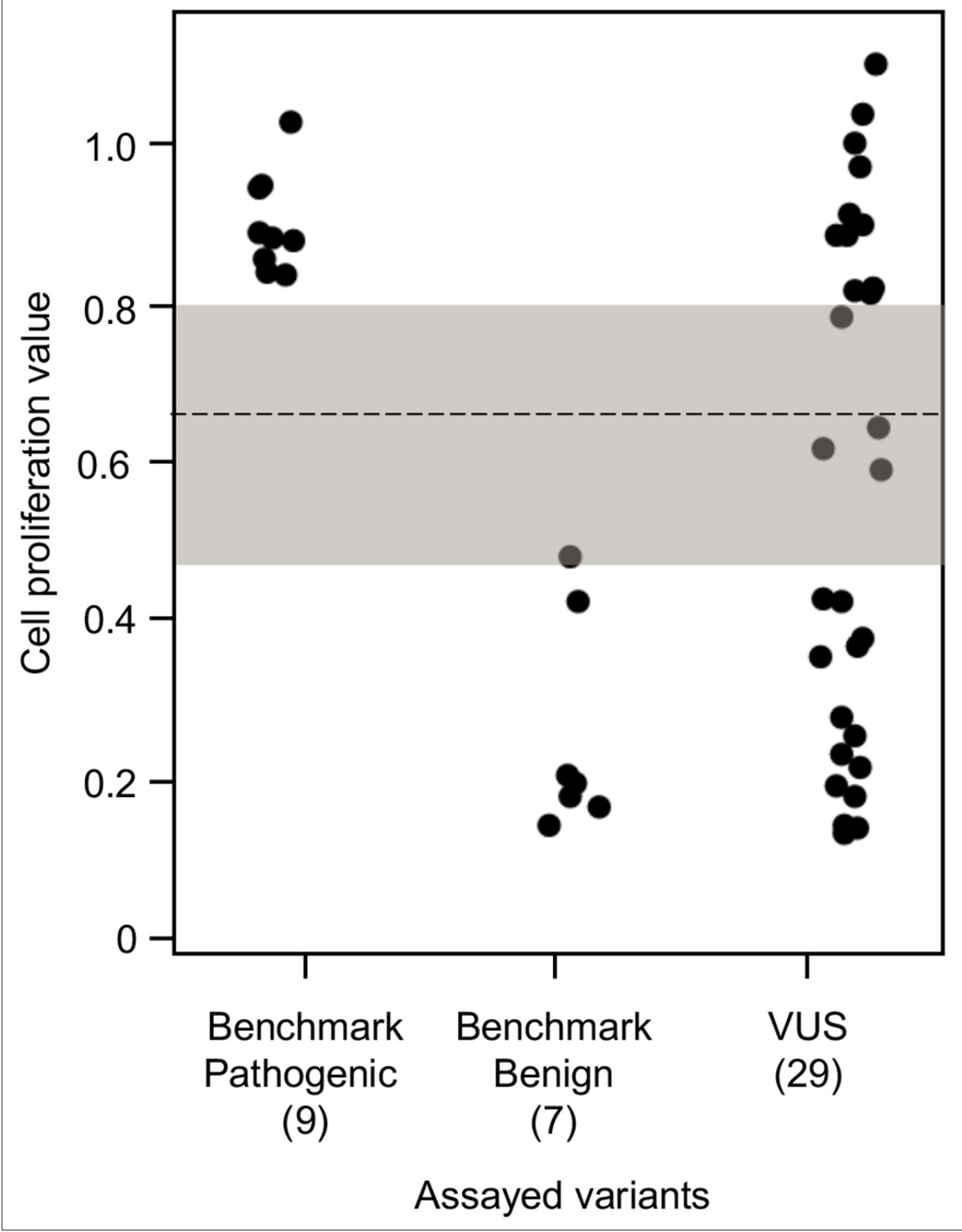

**Figure 3.** Functional characterization of *CDKN2A* variants. Functional characterization of 45 *CDKN2A* variants (9 pathogenic variants, 7 benign variants, 29 variants of uncertain significance [VUSs]) in PANC-1 cells. Cell proliferation values are shown as mean (n = 3) ± s.d. normalized to PANC-1 cell stably expressed with empty vector. Assayed variants with mean cell proliferation values: (1) above the gray shaded area (>0.81) were classified as functionally deleterious, (2) below the gray shaded area (<0.44) were classified as functionally benign, (3) in the gray area above the threshold dotted line (≥0.66 but

*Figure 3 continued on next page*

*Figure 3 continued*

≤0.81) were potentially functionally deleterious, and (4) in the gray area below the threshold dotted line (<0.66 but ≥0.44) were potentially functionally neutral. For raw data in this figure, please refer to *Figure 3—source data 1*.

The online version of this article includes the following source data and figure supplement(s) for figure 3:

**Source data 1.** Raw data in *Figure 3*.

**Figure supplement 1.** CDKN2A expression in human pancreatic ductal adenocarcinoma (PDAC) cells.

**Figure supplement 1—source data 1.** Original file of the full raw unedited CDKN2A western blot.

**Figure supplement 1—source data 2.** Original file of the full raw unedited vinculin western blot.

**Figure supplement 1—source data 3.** The uncropped western blots with the relevant CDKN2A and vinculin bands labelled.

**Figure supplement 2.** Functional characterization of *CDKN2A* variants in MIA PaCa-2 and AsPC-1 cells.

**Figure supplement 2—source data 1.** Raw data in *Figure 3—figure supplement 2*.

**Figure supplement 3.** Functional characterization of *CDKN2A* variants.

**Figure supplement 3—source data 1.** Raw data in *Figure 3—figure supplement 3*.

## Analysis of cell cycle progression

As reintroduction of CDKN2A into cancer cells lacking CDKN2A delays cell cycle progression, we determined the percent of variant-expressing PANC-1 cells in G1, G2/M, and S phases of the cell cycle (*Miller et al., 2011*). The percent of cells in G1 phase for benchmark pathogenic variants (47.2%, range: 43–50.9%) and VUSs with functionally deleterious or potentially functionally deleterious classifications (48.5%, range: 42.2–52.3%) were significantly lower than benchmark benign variants (63.1%, range: 58.4–67.1%) (p-value < 0.01, Student's t test) and VUSs with functionally neutral or potentially functionally neutral classifications (62.8%, range: 57.6–68.8%) (p-value < 0.01, Student's t test) (*Figure 4* and *Figure 4—figure supplements 1–4*). Similarly, the percent of cells in G2/M phase for benchmark pathogenic variants (42.4%, range: 37.7–49.2%) and VUSs with functionally deleterious or potentially functionally deleterious classifications (40.4%, range: 37.3–45.5%) were significantly lower than benchmark benign variants (25.7%, range: 19.3–35.1%) and VUSs with functionally neutral or potentially functionally neutral classifications (26.6%, range: 18.9–36.6%) (p-value < 0.01, Student's t test) (*Figure 4—figure supplement 5*). These data are consistent with the functional classifications from our cell proliferation assay.

## Comparison with previously reported functional data

Five of the 29 *CDKN2A* VUSs assayed had previously reported functional data available (*Supplementary file 3*). In agreement with our functional classification, p.Arg24Gln and p.Gly101Arg did not demonstrate deleterious effects in previously reported CDK4 binding and cell proliferation assays (*Jenkins et al., 2013*; *Kannengiesser et al., 2009*; *Miller et al., 2011*). p.Gly122Val previously demonstrated CDK6 binding similar to WT CDKN2A, binding to CDK4 was significantly reduced, and partially inhibit cell proliferation (*Yakobson et al., 2000*). In our study, p.Gly122Val partially inhibited cell proliferation and clustered with potentially functionally neutral variants. Furthermore, p.Gly67Arg and p.Glu69Gly were previously shown to result in a 40% reduction in CDK4 binding compared to WT CDKN2A (*Kannengiesser et al., 2009*). In our study, p.Gly67Arg and p.Glu69Gly clearly inhibited cell proliferation and clustered with functionally neutral variants.

## Prevalence of functionally deleterious *CDKN2A* VUS in patients with PDAC

As the assayed *CDKN2A* VUSs included variants identified in previously published studies of patients with PDAC, we next determined the prevalence of functionally deleterious VUSs, VUSs that could be reclassified as likely pathogenic variants, in separate cohorts of patients with PDAC. Ten (58.8%) of the 17 *CDKN2A* VUSs identified in a study of 638 patients with familial pancreatic cancer (*Roberts et al., 2016*) were found to be functionally deleterious in our assay and could be reclassified as likely pathogenic variants. In another study of 727 patients with a family history of PDAC, including patients that did not meet the criteria for classification as familial pancreatic cancer, 6 (46.2%) of 13 *CDKN2A* VUSs identified were functionally deleterious in our assay (*Zhen et al., 2015*). Similarly, in a study of 302 patients with a family history of PDAC, 4 of the 9 (44.4%) *CDKN2A* VUSs reported were

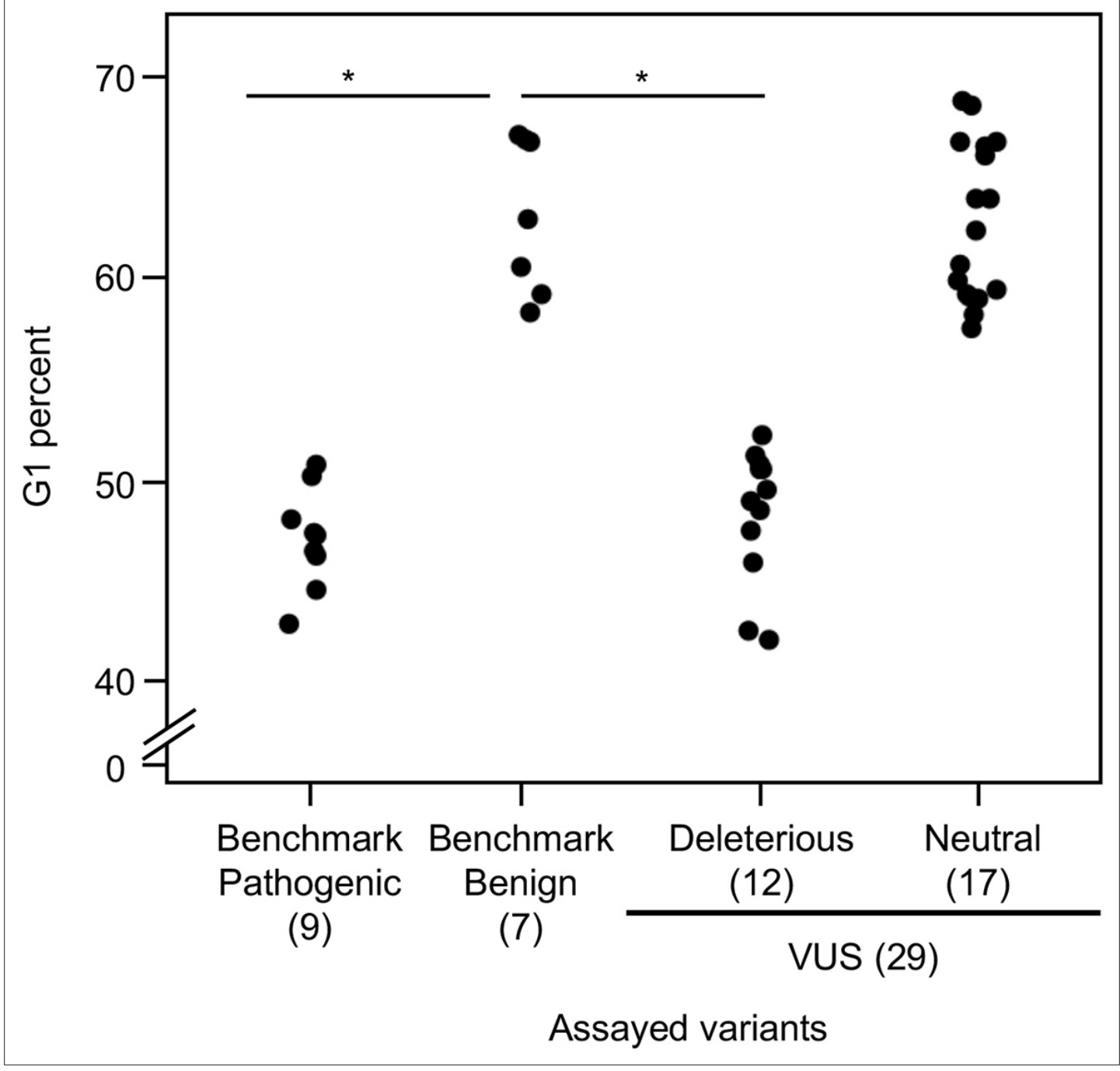

**Figure 4.** Cell cycle analysis of *CDKN2A* variants. G1 percent for 45 *CDKN2A* variants (9 pathogenic variants, 7 benign variants, 29 variants of uncertain significance [VUSs]) in PANC-1 cells. Significant inhibition of G1 arrest function of CDKN2A by benchmark pathogenic variants and functionally deleterious or potentially functionally deleterious VUSs. *p < 0.01. For raw data in this figure, please refer to *Figure 4* and *Table 1*.

The online version of this article includes the following source data and figure supplement(s) for figure 4:

**Source data 1.** Raw data in *Figure 4* and *Figure 4—figure supplement 5*.

**Figure supplement 1.** Cell cycle analysis of benchmark pathogenic *CDKN2A* variants.

**Figure supplement 2.** Cell cycle analysis of benchmark benign *CDKN2A* variants.

**Figure supplement 3.** Cell cycle analysis of functionally deleterious and potentially functionally deleterious *CDKN2A* variants of uncertain significance (VUSs).

**Figure supplement 4.** Cell cycle analysis of functionally neutral and potentially functionally neutral *CDKN2A* variants of uncertain significance (VUSs).

*Figure 4 continued on next page*

*Figure 4 continued*

**Figure supplement 5.** G2_M percent of *CDKN2A* variants expression cells.

**Figure supplement 5—source data 1.** Raw data in *Figure 4* and *Figure 4—figure supplement 5*.

functionally deleterious in our assay (*Chaffee et al., 2018*). Combining these latter two study results, there were 10 functionally deleterious *CDKN2A* variants that could be reclassified as likely pathogenic variants in 1029 patients with a family history of PDAC.

When considering patients with PDAC unselected for family history, in one study of 350 patients with PDAC (*McWilliams et al., 2018*), 3 of 8 (37.5%) VUSs identified were classified as functionally deleterious in our assay. On the other hand, in another study of 854 patients with PDAC unselected for family history, the one VUS identified was found to be functionally neutral (*Shindo et al., 2017*). Similarly, when combining these data, three functionally deleterious *CDKN2A* variants that could be reclassified as likely pathogenic variants were identified in 1204 patients with PDAC unselected for family history.

As we assayed all *CDKN2A* VUSs reported in these studies, our data indicate that the prevalence of likely pathogenic germline *CDKN2A* variants initially characterized as VUSs is 1.6% (95% CI: 0.8–2.9%) in patients with familial pancreatic cancer, 0.9% (95% CI: 0.5–1.8%) in patients with PDAC positive family history of PDAC (including patients that did not meet the criteria for classification as familial pancreatic cancer), and 0.1% (95% CI: 0.02–0.3%) in PDAC patients unselected for family history. When incorporating functionally deleterious VUSs we have reclassified as likely pathogenic variants with known pathogenic variants in *CDKN2A*, the cumulative prevalence of pathogenic and likely pathogenic *CDKN2A* variants in these studies increased from 2.5% to 4.1% (95% CI: 2.8–5.9%) in patients with familial pancreatic cancer, from 1.7% to 2.7% (95% CI: 1.9–3.9%) in patients with PDAC who had a positive for a family history of the disease (including patients that did not meet the criteria for classification as familial pancreatic cancer), and from 0.7% to 1.0% (95% CI: 0.6–1.7%) in PDAC patients unselected for family history.

## Discussion

Germline *CDKN2A* VUSs are identified in up to 4.3% patients with PDAC and are the cause of significant clinical uncertainty. Individuals with a *CDKN2A* VUS are not currently eligible for surveillance unless they meet other family history criteria. We used a cell proliferation assay to functionally characterize germline *CDKN2A* VUSs identified in patients with PDAC and/or reported in ClinVar and provide evidence to reclassify variants into clinically actionable strata.

We found that over 40% of *CDKN2A* variants previously classified as VUS were functionally deleterious in our assay and could be reclassified as likely pathogenic variants (12 of 29 *CDKN2A* VUSs; 41.4%). When considering *CDKN2A* VUS identified in patients with PDAC, our results suggest that the prevalence of *CDKN2A* VUSs that are functionally deleterious is up to 1.6% in patients with PDAC, depending on family history. It should be noted, however, that we characterized only missense *CDKN2A* VUS in our assay. Functional characterization of other types of *CDKN2A* VUS, such as synonymous variants, in-frame indels, and non-coding variants, would refine these estimates.

Furthermore, we determined the prevalence of germline *CDKN2A* VUSs that are functionally deleterious in patients with PDAC reported in previously published literature. Overlap of patient cohorts between these studies may affect estimates of prevalence. Regardless, we can conclude that functionally deleterious *CDKN2A* VUSs are likely to play a significant and previously unappreciated role in risk of PDAC, particularly for patients with a family history of the disease. Studies to define age-specific penetrance estimates for individuals with a pathogenic germline *CDKN2A* variant, including functionally deleterious variants previously classified as VUSs, are necessary and will inform genetic counselling, surveillance, and early detection efforts.

VUS functional characterizations were generally consistent when using PANC-1, MIA PaCA-2, and AsPC-1 cells in our cell proliferation assay. Specifically, no variant found to be functionally deleterious or potentially functionally deleterious when assayed with one cell line, had a functionally neutral or potentially functionally neutral classification when assayed with another cell line. Furthermore, PDAC cell lines had different genetic alterations and degrees of ploidy (*Sirivatanauksorn et al., 2001*;

*Yamato and Furukawa, 2001*) indicating that our functional classifications are robust to cell line-specific effects.

Consistent with our functional classifications, we found that benchmark pathogenic variants and VUSs classified as functionally deleterious or potentially functionally deleterious had a significantly lower percent of cells in G1 phase and a significantly higher percent of cells in G2/M phase of the cell cycle compared to either benchmark benign variants or VUSs classified as functionally benign or potentially functionally benign. These data suggest that expression of *CDKN2A* variants classified as pathogenic, functionally deleterious, or potentially functionally deleterious do not control progression through the cell cycle.

We were also able to compare our assay results for five *CDKN2A* VUSs with previously published data. In general, our assay results were consistent with previous reports. However, assay results for two VUSs, p.Gly67Arg and p.Glu69Gly, were inconsistent with previously reported partial reduction in CDK4 binding. There are several explanations for these observations. First, our assay is of broad cellular function, specifically cell proliferation, and therefore, may encompass functions other than CDK4 binding. Second, it is possible that overexpression of *CDKN2A* variants with partial loss-of-function may be obscured in our assay by non-physiological levels of expression. Future studies to correlate CDK4 binding with pathogenicity and to develop of functional assays that utilize endogenous promoters and enhancers would provide further clarity.

In conclusion, we functionally characterized and reclassified 29 *CDKN2A* VUSs identified in patients with PDAC or reported in ClinVar using a validated functional assay. We found that over 40% of *CDKN2A* VUS assayed were, in fact, functionally deleterious and as a result can be reclassified as likely pathogenic using ACMG guidelines. This finding may have significant implications for the management of patients with PDAC, and their relatives found to have a *CDKN2A* VUS.

## Materials and methods
### Cell lines
PANC-1 (catalog no. CRL-1469), MIA PaCa-2 (catalog no. CRL-1420), and AsPC-1 (catalog no. CRL-1682), and the human embryonic kidney cell line 293T (catalog no. CRL-3216) were purchased from American Type Culture Collection (Manassas, VA). PANC-1, MIA PaCa-2, and 293T cells were maintained in Dulbecco's modified Eagle's medium supplemented with 10% fetal bovine serum. AsPC-1 cells were maintained in RPMI-1640 supplemented with 10% fetal bovine serum. Cell line authentication and mycoplasma testing of PDAC cell lines was performed using GenePrint 10 System (Promega Corporation, Madison, WI; catalog no. B9510) and a PCR-based MycoDtect kit (Greiner Bio-One, Monroe, NC; catalog no. 463 060), respectively, by the Genetics Resource Core Facility of The Johns Hopkins University School of Medicine, Baltimore, MD.

### Western blot
Western blots were performed as previously described with the following modifications (*Tamura, 2018*): (1) anti-vinculin primary antibody was used at a 1:5000 dilution (Cell Signaling Technology, Beverly, MA; catalog no. 13901), (2) anti-CDKN2A primary antibody was used to detect p16[INK4A] at a 1:1000 dilution (Cell Signaling Technology, Beverly, MA; catalog no. 92803). Western blotting data and images presented are representative of at least three independent experiments.

### Plasmid construction
pHAGE-CDKN2A (Addgene, plasmid no. 116726) and pHAGE-CDKN2A-L78Hfs*41 (Addgene, plasmid no. 116222) created by Gordon Mills & Kenneth Scott (*Ng et al., 2018*), and pLJM1-Empty (Addgene, plasmid no. 91980) created by Joshua Mendell (*Golden et al., 2017*) were obtained from Addgene (Watertown, MA). *CDKN2A* and *CDKN2A*-L78Hfs*41 cDNAs were subcloned into the pLJM1-Empty from pHAGE-CDKN2A and pHAGE-CDKN2A-L78Hfs*41, respectively, using the Q5 High-Fidelity PCR kit (New England Biolabs, Ipswich, MA; catalog no. E0555S) as previously described (*Zhen et al., 2017*). Primers used to subclone *CDKN2A* and *CDKN2A*-L78Hfs*41 cDNAs are given in *Supplementary file 4*. Integration of *CDKN2A* cDNA was confirmed by restriction enzyme digest using EcoRI-HF (New England Biolabs, Ipswich, MA; catalog no. R3101S) and Sanger sequencing (Genewiz, Plainsfield, NJ). Restriction enzyme, EcoRI-HF digest was performed with CutSmart Buffer

(New England Biolabs, Ipswich, MA; catalog no. B7204S) at 37°C for 15 min. Expression constructs for each assayed *CDKN2A* variant were produced from the pLJM1-CDKN2A using the Q5 Site-Directed Mutagenesis kit (New England Biolabs, Ipswich, MA; catalog no. E0552). Primers used to generate each *CDKN2A* variant expression plasmids are given in *Supplementary file 4*. Integration of each *CDKN2A* VUS was confirmed using Sanger sequencing (Genewiz, Plainsfield, NJ). The manufacturer's protocol was followed unless otherwise specified.

## Lentivirus production and generation of CDKN2A expressing PANC-1 cells

pLJM1 lentiviral expression vectors harboring a *CDKN2A* cDNA (WT or variant) and the lentiviral packaging vectors, psPAX2 (Addgene, plasmid no. 12260) and pCMV-VSV-G (Addgene, plasmid no. 8454), created by Didier Trono and Bob Weinberg, respectively (*Stewart et al., 2003*), were transfected into 293T cells using Lipofectamine 3000 transfection reagent (Thermo Fisher Scientific, Waltham, MA; catalog no. L3000008). Media was collected at 24 and 48 hr and lentiviral particles were concentrated with Lenti-X Concentrator (Clontech, Mountain View, CA; catalog no. 631231). Manufacturer's protocols were used unless otherwise specified.

PANC-1, MIA PaCa-2, and AsPC-1 cells stably expressing WT CDKN2A (p16$^{INK4A}$) or selected variants were generated by lentiviral transduction. Briefly, $1 \times 10^5$ cells were cultured in media supplemented with 10 μg/ml polybrene and transduced with 10 TUs/ml lentivirus particles. Cells were then centrifuged at 1200× *g* for 1 hr. After 48 hr of culture at 37°C and 5% $CO_2$, transduced cells were selected by culture in media containing 3 μg/ml puromycin for PANC-1 cells and 1 μg/ml puromycin for MIA PaCa-2 and AsPC-1 cells for 7 days.

## Cell proliferation assay and functional characterization

$1 \times 10^5$ PANC-1, MIA PaCa-2, and AsPC-1 cells were seeded into culture (day 0) and cell numbers were counted using a TC20 Automated Cell Counter (Bio-Rad Laboratories, Hercules, CA, catalog no. 1450102) on day 7 for MIAPaCa-2 cells, and day 14 for PANC-1 and AsPC-1 cells. Cell numbers were normalized to empty vector transduced cells to give a relative cell proliferation value. Assays to determine the functional effect of *CDKN2A* variants were repeated in triplicate. Mean cell proliferation value and standard deviation (s.d.) were calculated.

*CDKN2A* VUSs were functionally characterized into four categories: (1) deleterious, (2) potentially deleterious, (3) potentially neutral, and (4) neutral, based on the results of the cell proliferation assay using previously described methodology (*Bouvet et al., 2019*). Briefly, functionally deleterious was defined as cell proliferation values greater than the Z-score 95% CI upper limit calculated from benchmark pathogenic variants and benign variants. Functionally neutral was defined as proliferation values less than the Z-score 95% CI lower limit. Assayed variants with cell proliferation values between the thresholds for functionally deleterious and functionally neutral were characterized as potentially functionally deleterious or potentially functionally neutral if they had proliferation values above or below a cutoff determined by the ROC curve and Youden index.

## Cell cycle analysis

PANC-1 cells were incubated with 10 nM EdU for 2 hr, then harvested by trypsinization, washed, fixed, and stained using the Click-iT EdU Alexa Fluor 488 Flow Cytometry Assay Kit (Invitrogen, Carlsbad, CA, catalog no. C10425) and the FxCycle Violet Ready Flow Reagent (Invitrogen, catalog no. R37166) according to the manufacturer's protocols. Flow cytometry was performed on Cytek Aurora Flow Cytometry System (Cytek Biosciences, Fremont, CA) and the data analyzed to determine the percentage of cells in each phase of the cell cycle using the FlowJo software (BD Biosciences, San Jose, CA).

## Statistical analyses

Statistical analyses were performed using JMP v.11 (SAS, Cary, NC) and Prism v.6 (GraphPad, San Diego, CA). Comparison of means used the Student's t test. CIs for the prevalence of functionally deleterious *CDKN2A* variants were calculated using the modified Wald Method. A p-value less than 0.05 considered statistically significant.

## Acknowledgements

The authors would like to thank to Dr Xiaoling Zhang at The Johns Hopkins University School of Medicine Ross Flow Cytometry Core Facility for technical assistance. Data used in this article were obtained from the Familial Pancreatic Cancer Genome Sequencing Project (FPC-GSP; https://www.familialpancreaticcancer.org). The FPC-GSP investigators contributed to the design and implementation of FPC-GSP and/or provided data. They did not participate in analysis or writing of this report. The FPC-GSP project was generously supported by Dennis Troper and Susan Wojcicki, the Lustgarten Foundation for Pancreatic Cancer Research, the Sol Goldman Pancreatic Cancer Research Center, the Virginia and DK Ludwig Fund for Cancer Research, the Michael Rolfe Foundation, the Joseph C Monastra Foundation, the Gerald O Mann Charitable Foundation, the Ladies Auxiliary to the Veterans of Foreign Wars, the friends and family of Roger L Kerns Sr, the Weston Garfield Foundation, the NIH Specialized Programs of Research Excellence P30CA006973 and P50CA102701, and NIH grants K99-CA190889, R01-CA57345, R01-CA97075, R01-CA154823, and R01-DK060694.

## Additional information

### Group author details

**The Familial Pancreatic Cancer Genome Sequencing Project**

**Randall Brand**: Department of Medicine, University of Pittsburgh, Pittsburgh, United States; **Michele L Cote**: Karmanos Cancer Institute, Wayne State University School of Medicine, Detroit, United States; **Mengmeng Du**: Department of Epidemiology and Biostatistics, Memorial Sloan Kettering Cancer Center, New York, United States; **James R Eshleman**: ; **Steven Gallinger**: Ontario Institute for Cancer Research, Toronto, Canada; **Michael Goggins**: ; **Ralph H Hruban**: ; **Alison P Klein**: ; **Robert C Kurtz**: Department of Medicine, Memorial Sloan Kettering Cancer Center, New York, United States; **Gloria M Petersen**: Department of Health Sciences Research, Mayo Clinic, Rochester, United States; **Nicholas J Roberts**: ; **Anil K Rustgi**: Herbert Irving Comprehensive Cancer Center, Columbia University Irving Medical Center, New York, United States; **Ann G Schwartz**: Karmanos Cancer Institute, Wayne State University School of Medicine, Detroit, United States; **Elena M Stoffel**: Department of Internal Medicine, University of Michigan, Ann Arbor, United States; **Sapna Syngal**: Dana-Farber Cancer Institute and Harvard Medical School, Boston, United States; **George Zogopoulos**: The Research Institute of the McGill University Health Centre, Quebec, Canada

### Competing interests

Ralph H Hruban: RHH has the right to receive royalty payments from Thrive Earlier Diagnosis for the GNAS in pancreatic cysts invention in a relationship overseen by Johns Hopkins University. The other authors declare that no competing interests exist.

### Funding

| Funder | Grant reference number | Author |
| --- | --- | --- |
| The Sol Goldman Pancreatic Cancer Research Center | Pilot Grant | Nicholas Jason Roberts |
| The Rolfe Pancreatic Cancer Foundation | | Nicholas Jason Roberts |
| National Institutes of Health | P50 CA622924 | Alison P Klein |
| The Japanese Society of Gastroenterology | | Hirokazu Kimura |
| The Japan Society for Promotion of Science | | Hirokazu Kimura |
| The Joseph C Monastra Foundation | | Nicholas Jason Roberts |

| Funder | Grant reference number | Author |
| --- | --- | --- |
| The Geral O Mann Foundation | | Nicholas Jason Roberts |
| Art Creates Cures Foundation | | Nicholas Jason Roberts |
| Susan Wojcicki and Denis Troper | | Nicholas Jason Roberts |
| National Institutes of Health | S10 OD026859 | Nicholas Jason Roberts |

The funders had no role in study design, data collection and interpretation, or the decision to submit the work for publication.

## Author contributions

Hirokazu Kimura, Conceptualization, Formal analysis, Funding acquisition, Investigation, Methodology, Project administration, Resources, Supervision, Validation, Visualization, Writing – original draft, Writing – review and editing; Raymond M Paranal, Conceptualization, Formal analysis, Investigation, Methodology, Resources, Validation, Visualization, Writing – original draft, Writing – review and editing; Neha Nanda, Validation; Laura D Wood, Investigation, Resources, Validation, Writing – review and editing; James R Eshleman, Ralph H Hruban, Michael G Goggins, Alison P Klein, The Familial Pancreatic Cancer Genome Sequencing Project, Resources, Writing – review and editing; Nicholas J Roberts, Conceptualization, Formal analysis, Funding acquisition, Methodology, Project administration, Resources, Supervision, Validation, Visualization, Writing – original draft, Writing – review and editing

## Author ORCIDs

Nicholas J Roberts  http://orcid.org/0000-0002-8709-0664

## Decision letter and Author response

Decision letter https://doi.org/10.7554/eLife.71137.sa1
Author response https://doi.org/10.7554/eLife.71137.sa2

# Additional files

## Supplementary files

- Transparent reporting form
- Supplementary file 1. CDKN2A variants assayed with genomic, transcript, and protein change.
- Supplementary file 2. Functional characterization.
- Supplementary file 3. Previously reported functional data for assayed CDKN2A variants.
- Supplementary file 4. Primer sequences used for subconing and site directed mutagenesis.

## Data availability

All data generated or analysed during this study are included in the manuscript, supporting files, and as Source data and Source code.

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
