## [Decision Letter]

**Decision letter after peer review:**

Thank you for resubmitting your work entitled "Functional CDKN2A assay identifies frequent deleterious alleles misclassified as variants of unknown significance" for further consideration by *eLife*. Your article has been reviewed by 2 reviewers, one of whom is member of our Board of Reviewing Editors and the evaluation has been overseen by Wafik El-Deiry as the Senior Editor. The reviewers have opted to remain anonymous.

The manuscript has been improved but there are some remaining issues that need to be addressed, as outlined below:

*Reviewer #1:*

The investigators postulated that some variants of uncertain significance (VUS) of CDKN2A have been misclassified, and that some proportion may be in fact be deleterious and possibly pathogenic in carcinogenesis of pancreatic ductal adenocarcinoma (PDAC). in vitro analysis of cell proliferation of 29 known VUSs was performed, and the authors report that >40% of these variants are functionally deleterious.

CDKN2A encodes a tumor suppressor as well as an important regulator of progression through the cell cycle. It has been extensively studied in epithelial carcinomas, including pancreatic ductal adenocarcinomas (PDAC). The concept that a significant number of previously reported variants of this gene may not be as benign has important implications for understanding etiology and downstream molecular effects and tumoral progression of this cancer, especially in the albeit relatively small number of patients who harbor germline variants that may be important in driving pancreatic cancer development. The study is well-written and leverages an established and validated in vitro cell proliferation assay from this lab.

1. The information redesignating 40% of the VUS as functionally deleterious and possibly pathogenic is applied to external datasets. Did the investigators gain access to the raw data from the authors of the cited published works, or were the reassignments based on available information that was published? It would be important to access and confirm the raw data.

2. Application to an internal dataset of tumor samples would affirm the conclusions made from the in vitro validation steps.

3. The use of only a single PDAC cell line is a limitation that tempers the conclusions of the study [PANC-1 is a hypertriploid cell line; what is the effect in relation to CDKN2A?].The in vitro analysis should be performed on at least three cell lines to control for other differences between cell lines. This work would strengthen the argument, as the premise of the study is to extend the finding to in vivo/human PDAC, with implications that the reassigned germline VUSs are more pathogenic in humans than previously thought.

*Reviewer #2:*

The study provides evidence of functionality of mutations present in pancreatic cancer, a dismal condition predicted be the second cause of cancer death by 2030. Specifically, the authors showed that mutations in CDKN2A associated with increased risk of developing pancreatic cancer have lost the ability to control cell growth.

The authors have functionally characterized CDKN2A mutations present in familial pancreatic cancer cases. The methodology used is for the most adequate to address the main goal of the study. Some of the deficiencies/weaknesses are associated with limitations of the pancreatic cancer model/methodologies and lack of side by side comparison with other known factors associated with increased risk of the disease as well as clinical correlations of these mutations.

To fully support the authors conclusions, the authors should

1) The Panc1 results should be repeated in two additional lines to avoid any cell line bias effect.

2) The cell growth analysis should be confirmed using a cell cycle analysis to define if the mutants lack the ability to control G1 progression.

3) The study should include side-by-side comparison with other know p16 mutations from familial cases.

---

## [Author Response]

Reviewer #1:The investigators postulated that some variants of uncertain significance (VUS) of CDKN2A have been misclassified, and that some proportion may be in fact be deleterious and possibly pathogenic in carcinogenesis of pancreatic ductal adenocarcinoma (PDAC). in vitro analysis of cell proliferation of 29 known VUSs was performed, and the authors report that >40% of these variants are functionally deleterious.CDKN2A encodes a tumor suppressor as well as an important regulator of progression through the cell cycle. It has been extensively studied in epithelial carcinomas, including pancreatic ductal adenocarcinomas (PDAC). The concept that a significant number of previously reported variants of this gene may not be as benign has important implications for understanding etiology and downstream molecular effects and tumoral progression of this cancer, especially in the albeit relatively small number of patients who harbor germline variants that may be important in driving pancreatic cancer development. The study is well-written and leverages an established and validated in vitro cell proliferation assay from this lab.1. The information redesignating 40% of the VUS as functionally deleterious and possibly pathogenic is applied to external datasets. Did the investigators gain access to the raw data from the authors of the cited published works, or were the reassignments based on available information that was published? It would be important to access and confirm the raw data.

We are happy to clarify. Several authors of this manuscript were also authors of the cited published works used to identify VUS for assay. These authors were extensively involved in the analysis and interpretation of these variant data. The cited studies included research next-generation sequencing (whole genome and gene-panel), Sanger sequencing, and CLIA approved clinical genetic testing that have high-confidence for single base substitution calls. As such, we are confident in the VUS calls used in this manuscript.

2. Application to an internal dataset of tumor samples would affirm the conclusions made from the in vitro validation steps.

We thank the reviewer for their suggestion. We agree that analysis of tumor samples might provide additional evidence to validate functional characterizations. Unfortunately, only ~10% of patients with pancreatic ductal adenocarcinoma undergo resection and have tissue available for correlative studies. We were unable to identify any tumor tissue for patients with functionally deleterious or potentially functionally deleterious VUS where additional studies would be informative, for example, somatic *CDKN2A* loss-of-heterozygosity with retention of the deleterious variant (supporting evidence) or somatic loss of the deleterious variant (refuting evidence). However, even without tumor samples, we present additional experimental evidence to support our functional characterizations (see below).

3. The use of only a single PDAC cell line is a limitation that tempers the conclusions of the study [PANC-1 is a hypertriploid cell line; what is the effect in relation to CDKN2A?].The in vitro analysis should be performed on at least three cell lines to control for other differences between cell lines. This work would strengthen the argument, as the premise of the study is to extend the finding to in vivo/human PDAC, with implications that the reassigned germline VUSs are more pathogenic in humans than previously thought.

We are happy to provide the additional experimental data requested by the reviewer. We have now assayed selected *CDKN2A* variants (pathogenic, benign, and VUS) in two additional cell lines (MIA PaCa-2 and AsPC-1). Reassuringly, no VUS with a functionally deleterious or potentially functionally deleterious classification in one cell line had a functionally neutral or potentially functionally neutral classification in the other cell lines tested. Furthermore, cell cycle analysis to determine the distribution of cells in each stage of the cell cycle provides support for our functional classifications. These additional data indicate that our functional classifications are robust to cell lines specific effects included ploidy and genetic background.

Reviewer #2:The study provides evidence of functionality of mutations present in pancreatic cancer, a dismal condition predicted be the second cause of cancer death by 2030. Specifically, the authors showed that mutations in CDK2A associated with increased risk of developing pancreatic cancer have lost the ability to control cell growth.The authors have functionally characterized CDK2A mutations present in familial pancreatic cancer cases. The methodology used is for the most adequate to address the main goal of the study. Some of the deficiencies/weaknesses are associated with limitations of the pancreatic cancer model/methodologies and lack of side by side comparison with other known factors associated with increased risk of the disease as well as clinical correlations of these mutations.To fully support the authors conclusions, the authors should1) The Panc1 results should be repeated in two additional lines to avoid any cell line bias effect.

We are happy to provide the additional experimental data requested by the reviewer. We have now assayed selected *CDKN2A* variants (pathogenic, benign, and VUS) in two additional cell lines (MIA PaCa-2 and AsPC-1). Reassuringly, no VUS with a functionally deleterious or potentially functionally deleterious classification in one cell line had a functionally neutral or potentially functionally neutral classification in the other cell lines tested.

2) The cell growth analysis should be confirmed using a cell cycle analysis to define if the mutants lack the ability to control G1 progression.

We thank the reviewer for this suggestion. We have completed cell cycle analysis to determine the distribution of cells in each stage of the cell cycle for each assayed variant in PANC-1 cells. These data provide clear support for our functional classifications and indicate that expression of pathogenic or functionally deleterious variants do not inhibit cell cycle progression. Specifically, pathogenic and functionally deleterious variants had a lower percentage of cells in G1 phase and a higher percentage of cells in G2/M phase compared to benign and functionally neutral variants.

3) The study should include side-by-side comparison with other know p16 mutations from familial cases.

We are grateful for the opportunity to clarify. The majority of pathogenic germline variants tested in our assay were either identified in patients with familial pancreatic cancer (5 of 9 variants) or in patients with pancreatic adenocarcinoma (2 of 9). Furthermore, these variants were classified as pathogenic using American College of Medical Genetics variant interpretation guidelines. As such, our assay has broad applicability to patients with pancreatic cancer.